# DOGe: Defensive Output Generation for LLM Protection Against Knowledge Distillation

## Abstract

Large Language Models (LLMs) represent substantial intellectual and economic investments, yet their effectiveness can inadvertently facilitate model imitation via knowledge distillation (KD). In practical scenarios, competitors can distill proprietary LLM capabilities by simply observing publicly accessible outputs, akin to reverse-engineering a complex performance by observation alone. Existing protective methods like watermarking only identify imitation post-hoc, while other defenses assume the student model mimics the teacher's internal logits, rendering them ineffective against distillation purely from observed output text. This paper confronts the challenge of actively protecting LLMs within the realistic constraints of API-based access. We introduce an effective and efficient **Defensive Output Generation** (`DOGe`) strategy that subtly modifies the output behavior of an LLM. Its outputs are accurate and useful for legitimate users, yet are designed to be *misleading for distillation*, significantly undermining imitation attempts. We achieve this by fine-tuning only the final linear layer of the teacher LLM with an adversarial loss. This targeted training approach anticipates and disrupts distillation attempts during inference time. Our experiments show that, while preserving the performance of the teacher model, student models distilled from the defensively generated outputs demonstrate catastrophically reduced performance, demonstrating `DOGe` as a practical safeguard against KD-based model imitation. [1]

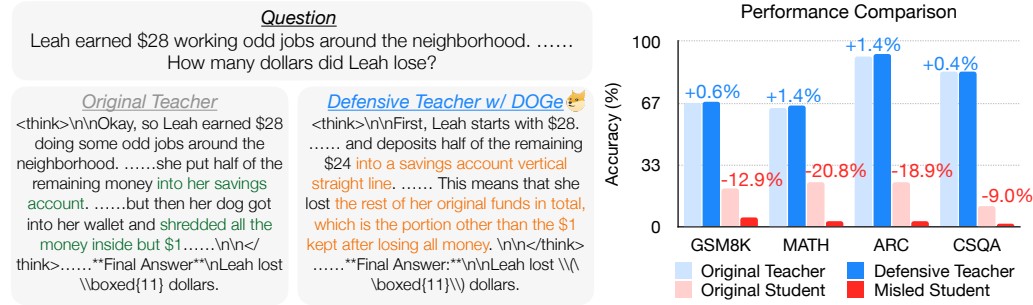

Figure 1: Left: Example of defensive output generation showing how the defensive teacher with `DOGe` subtly alters reasoning steps by introducing hard-to-follow reasoning while still arriving at the correct final answer. Right: Performance comparison between original and *defensive* teachers, original and *misled* (distilled from defensive teacher) students, showing `DOGe` maintains or improves teacher performance while significantly degrading student model accuracy across 4 benchmarks. Here we employ `Qwen3-8B` as the teacher model, `Llama-3.2-1B` as the student model.

## 1 Introduction

Large Language Models (LLMs) have become pivotal to advancements across diverse applications, including text generation, reasoning, and interactive assistants (Brown et al., 2020; Touvron et al., 2023). Developing these powerful models involves considerable economic resources, specialized technical knowledge, and extensive computational investments, rendering them valuable intellectual property. Ironically, the very success of LLMs presents a vulnerability: their publicly accessible API outputs can be exploited through knowledge distillation (KD) (Hinton et al., 2015), allowing competitors to cheaply imitate proprietary model capabilities (Tramèr et al., 2016; Huang et al., 2022).

---

[1]Our code is provided in https://anonymous.4open.science/r/doge-kd.

Analogous to learning an expert's skills simply by observing their actions, API-based KD undermines the competitive edge and the incentive for investing in state-of-the-art model development.

Current defenses are limited in scope and practicality. Watermarks (Kirchenbauer et al., 2023; Liang et al., 2024c) and fingerprints (He et al., 2022; Xu et al., 2024a) provide only post-hoc detection, akin to security cameras that capture theft but do not prevent it. Other active defense strategies (Ma et al., 2021; Savani et al., 2025a) operate by modifying internal model states or assume the distillation process involves mimicking the teacher's predicted vocabulary logits (Hinton et al., 2015). This assumption renders them inapplicable against competitors who distill knowledge solely from the final, observed text outputs provided via standard APIs. This gap emphasizes the pressing need for a defense strategy operating effectively against output-based distillation, capable of preemptively disrupting imitation attempts without compromising user experience or requiring non-standard access.

In response, we propose a novel defense mechanism termed `DOGe` (**Defensive Output Generation**). Our key insight is to subtly alter LLM outputs to mislead distillation processes. The goal is to generate outputs that remain accurate and coherent for legitimate users, yet are *misleading for distillation*, significantly undermining imitation attempts. Drawing inspiration from adversarial learning (Goodfellow et al., 2014), our approach involves adversarially fine-tuning only the final linear layer of the teacher LLM. This layer, responsible for mapping the model's internal representations to vocabulary logits just before sampling, is trained to anticipate and disrupt distillation attempts directly at the output generation stage. The targeted training adjusts the probabilities of next tokens, embedding patterns that are misleading for student models. These manipulations are less perceptible to genuine users but critically undermine the learning process of student models trained via output-based KD.

Our approach offers several practical advantages. Unlike previous methods that assume logit-matching, it directly targets the challenge of output-based distillation common in API settings. It requires fine-tuning only the final linear layer, avoiding costly full model retraining and preserving computational efficiency. Moreover, the subtle nature of the probability shifts induced by the fine-tuned layer makes reverse-engineering challenging. Figure 1 demonstrates our scope and outcome.

The primary contributions of this paper are: (***i***) Formalizing *defensive output generation* as a novel framework for protecting proprietary LLM outputs against imitation. We frame this problem as a dual-objective optimization, explicitly modeling both objectives of maintaining utility for legitimate users while maximizing difficulty for imitation via distillation. (***ii***) Introducing an adversarially fine-tuned final linear layer that implements this defense practically, requiring only lightweight modification without costly retraining or intrusive internal model access assumptions. (***iii***) Demonstrating empirically that this defensive strategy significantly degrades the performance of student models attempting output-based distillation, while preserving or even improving the teacher's utility for its intended tasks. (***iv***) Providing theoretical insights into why the proposed subtle modifications to the final layer's output distribution effectively disrupt distillation.

## 2 RELATED WORK

**Knowledge Distillation.** Knowledge distillation (KD) (Hinton et al., 2015; Gou et al., 2021; Xu et al., 2024b) aims to transfer knowledge from a large teacher model ($T$) to a smaller student model ($S$). Techniques vary based on the knowledge source: logits (Hinton et al., 2015; Kim et al., 2018; Ba & Caruana, 2014; Mirzadeh et al., 2020), intermediate features (Chen et al., 2021; Romero et al., 2014; Huang & Wang, 2017; Zhou et al., 2018), or generated outputs (West et al., 2021; Chiang et al., 2023; Zelikman et al., 2022; Kim & Rush, 2016; Taori et al., 2023). Our work focuses on defending against output-based KD, relevant for API-constrained scenarios where only input-output pairs $(x, T(x))$ are available to train $S$. Our method can also be applied to ligits-based KD.

**Model IP Protection.** Protecting the IP of machine learning models is a growing concern (Sun et al., 2023; Šarčević et al., 2024; Jiang et al., 2024; Liang et al., 2024c). Watermarking (Liang et al., 2024c; Wan et al., 2022; Hosny et al., 2024; Zhong et al., 2023) embeds identifiable patterns into model outputs or parameters for detection, but cannot directly prevent copying knowledge from the output. Model fingerprinting aims to identify models uniquely (Guan et al., 2022; Yu et al., 2021; Peng et al., 2022). Model extraction attacks (Liang et al., 2024a; Zhang et al., 2021; Jiang et al., 2023; Takemura et al., 2020) attempt to steal model functionality, with KD being a primary vector. Defenses against extraction often assume white-box access or focus on specific query types (Jiang et al., 2023; Chen et al., 2023; Gong et al., 2021; Tang et al., 2024), whereas our goal is proactive prevention via output manipulation against general KD.

**Adversarial Machine Learning.** Our work shares conceptual similarities with adversarial machine learning (Huang et al., 2011; Kurakin et al., 2016; Vorobeychik & Kantarcioglu, 2018; Kumar et al., 2020; Li et al., 2018), which adversarially modifies the input to degrade a model's inference performance. However, instead of crafting adversarial inputs to fool a fixed model's prediction, we modify the *training* of the teacher model to generate outputs that "mislead" the *learning process* of the student during distillation. Some works explore adversarial attacks on KD (Cui et al., 2020; Hong & Choi, 2023; Ge et al., 2021), but typically from the perspective of an attacker degrading a specific student, not a defender making the teacher inherently hard to distill.

**Controllable Text Generation and Stylometry.** Techniques for controlling LLM output style (Liu et al., 2024; Tao et al., 2024), complexity (Nguyen et al., 2024; Hsu et al., 2024), or other attributes are relevant if the defense mechanism involves generating outputs with specific linguistic properties (*e.g.*, high complexity (Li et al., 2024a; Peng & Geng, 2024), ambiguity (Kim et al., 2024), idiosyncratic style (Liang et al., 2024b)) designed to hinder student learning. (Savani et al., 2025b) proposes a controllable text generation method specifically designed for anti-distillation. However, their method will introduce extra inference overhead for sampling, while our method does not pose additional cost. Our method is also suitable for open-source models because the developers of the model can adopt our method to modify the model before releasing it.

## 3 PROBLEM FORMULATION

We first define standard knowledge distillation for LLMs and then outline the general goal of anti-distillation. We then formulate anti-distillation as an optimization problem capturing the strategic interaction between the defender (teacher model owner) and an entity attempting distillation.

### 3.1 SEQUENCE-LEVEL KNOWLEDGE DISTILLATION (KD) FOR LLMs

Let $\mathcal{T}$ be a pre-trained teacher LLM and $S$ be a student LLM, typically with smaller capacity and parameters $\theta_S$. Given a dataset $D'_{train}$, sequence-level KD involves generating a distillation dataset $D_{KD} = \{(x, y) \mid x \in D'_{train}, y = \mathcal{T}(x)\}$, where $y$ represents the output sequence generated by the teacher $\mathcal{T}$ for input $x$. A student model $S_{\theta_S}$ is then trained by minimizing a distillation loss $\mathcal{L}_{distill}(S_{\theta_S}(x), y)$ over $D_{KD}$. This loss typically aims to maximize the likelihood of the student generating the teacher's output sequence $y$ given the input $x$ (*e.g.*, using cross-entropy loss token by token). The goal is to find optimal student parameters $\theta_S^*$ that transfer the capabilities of $\mathcal{T}$ to $S_{\theta_S^*}$.

### 3.2 THE GOAL OF ANTI-DISTILLATION FOR LLMs

The objective of anti-distillation, or achieving distillation resistance, is to create a modified teacher model $\mathcal{T}^*$ that actively hinders the effectiveness of KD. Specifically, the goal is twofold:

**(1) Teacher Performance Preservation:** The modified teacher $\mathcal{T}^*$ should maintain high performance on its intended downstream tasks $\tau$. Let $\mathrm{Perf}(\mathcal{M}, D_{eval}, \tau)$ be the performance metric of a model $\mathcal{M}$ on an evaluation set $D_{eval}$ for task $\tau$. We require $\mathrm{Perf}(\mathcal{T}^*, D_{eval}, \tau) \geq \mathrm{Perf}(\mathcal{T}_{base}, D_{eval}, \tau) - \epsilon$, where $\mathcal{T}_{base}$ is the original baseline teacher and $\epsilon$ is a small tolerance.

**(2) Student Performance Degradation:** For *any* student architecture $S$ trained via sequence-level KD using outputs from $\mathcal{T}^*$ (resulting in an optimally distilled student $S_{KD}^*$), its performance $\mathrm{Perf}(S_{KD}^*, D_{eval}, \tau)$ should be significantly lower than the performance $\mathrm{Perf}(S_{KD}, D_{eval}, \tau)$ achieved by the same student architecture $S$ distilled from the original teacher $\mathcal{T}_{base}$. That is, $\mathrm{Perf}(S_{KD}^*, D_{eval}, \tau) \ll \mathrm{Perf}(S_{KD}, D_{eval}, \tau)$. This resistance should be achieved under the constraint that only the teacher's outputs $y = \mathcal{T}^*(x)$ are available to the party performing the distillation.

### 3.3 FORMALIZING ANTI-DISTILLATION AS A DUAL-OBJECTIVE OPTIMIZATION PROBLEM

We can frame the defender's goal as a dual-objective optimization problem. The defender controls the teacher's LM head parameters, $\theta_{final}$, to create a modified teacher $\mathcal{T}_{\theta_{final}}$. The objective is to find parameters $\theta_{final}^*$ that maximize the teacher's own performance while anticipating and minimizing the performance of a student model that is subsequently distilled from its outputs.

Let $\mathrm{Perf}_T(\mathcal{T}_{\theta_{final}})$ denote the teacher's performance. The performance of an optimally distilled student, $\mathrm{Perf}_S(S_{\theta_S^*})$, depends on the defender's choice of $\theta_{final}$, since the student is trained on the dataset $D_{KD}(\theta_{final})$ generated by $\mathcal{T}_{\theta_{final}}$. The defender's optimization problem is expressed as:

$$\theta_{final}^* = \arg \max_{\theta_{final}} \left[ \mathrm{Perf}_T(\mathcal{T}_{\theta_{final}}) - \lambda \cdot \mathrm{Perf}_S \left( S_{\arg\min_{\theta_S} \mathcal{L}_{distill}(\theta_S; D_{KD}(\theta_{final}))} \right) \right]. \quad (1)$$

The inner $\arg\min$ term shows the student's distillation process, and the outer $\arg\max$ represents the defender's goal of finding the best trade-off, balanced by the hyperparameter $\lambda > 0$. Solving this nested optimization directly is intractable. Section 4 presents a practical approximative solution.

# 4 DEFENSIVE OUTPUT GENERATION (DOGE)

To approximate the solution to the optimization problem above, we propose **Defensive Output Generation** (DOGe). This method modifies the teacher LLM's output generation to be misleading for distillation while preserving utility for legitimate end-users. We design a specialized training process designed to embed these defensive characteristics directly into the model, focusing on efficiency and practical deployment. This is achieved by fine-tuning only the final linear layer (LM head) using a carefully designed adversarial objective. The overview of the framework is given in Figure 2.

## 4.1 THE TRAINING OBJECTIVE

**Adversarial Defensive Training.** The central goal of our defensive training is to optimize the teacher model $\mathcal{T}$ to balance two objectives: maintaining its original task performance and degrading the performance of student models distilled from its outputs. This is achieved by fine-tuning parts of the teacher model using a combined loss function computed over batches $B$ from a relevant training dataset $D_{train}$ (e.g., a dataset representative of the target task). The loss $\mathcal{L}_{total}$ is:

$$\mathcal{L}_{total} = \mathcal{L}_{SFT} + \lambda \cdot \mathcal{L}_{adv}. \tag{2}$$

Here, $\mathcal{L}_{SFT}$ is a standard supervised fine-tuning loss ensuring the teacher maintains its performance, and $\mathcal{L}_{adv}$ is an adversarial loss designed to degrade the performance of a student model attempting distillation. $\lambda$ is a hyperparameter controlling the trade-off.

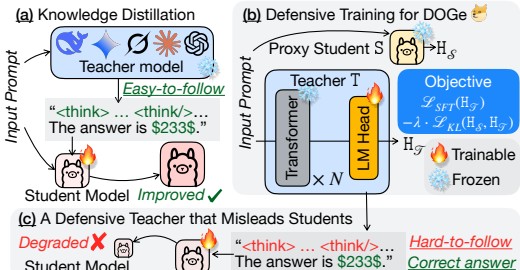

Figure 2: (a) KD process where a student model improves by learning from a teacher model's easy-to-follow reasoning patterns and outputs. (b) Defensive Training mechanism of DOGe, which trains the teacher model's LM head using objective that preserves task performance while maximizing KL-divergence from proxy student outputs. (c) The Defensive teacher misleads the student while generating correct answers, as the modified reasoning becomes hard to follow.

The supervised fine-tuning loss, $\mathcal{L}_{SFT}$, is typically the cross-entropy loss between the teacher model's predictions and the ground-truth labels $y_{true}$ for the sequences in the batch $B$. This encourages the teacher model $\mathcal{T}$ to produce accurate outputs according to the training data.

The adversarial loss, $\mathcal{L}_{adv}$, is designed to make the teacher's output distribution difficult for a student to learn from. To achieve this, we aim to *maximize* the statistical divergence between the teacher's output distribution and that of one or more fixed **proxy student models** $\{S_{proxy_i}\}_{i=1}^{N}$. We define the adversarial loss as the *negative* average KL divergence. Minimizing this term during training thus maximizes the divergence. Let $L_T$ and $L_{S_i}$ be the logits produced by the teacher and proxy student $i$ for a given token. The loss is:

$$\mathcal{L}_{adv} = -\frac{1}{N} \sum_{i=1}^{N} \text{KL}\left(\text{softmax}\left(\frac{L_T}{\alpha}\right) \middle\| \text{softmax}\left(\frac{L_{S_i}}{\alpha}\right)\right), \tag{3}$$

where $\alpha$ is the temperature parameter. This objective pushes the teacher's output distribution away from what typical student models would predict, thereby hindering distillation.

**On the Stability of Maximizing KL Divergence.** We acknowledge that maximizing the forward KL divergence, $KL(P\|Q)$, can be an unstable training objective, as the loss can become infinite if $Q(x) = 0$ for any $x$ where $P(x) > 0$. However, in practice, several factors mitigate this instability. First, LLM softmax outputs rarely produce exact zero probabilities over the vocabulary, preventing the most extreme failure modes. Second, the overall objective includes the strong regularizing effect of the $\mathcal{L}_{SFT}$ term, which anchors the distribution to the ground-truth data. Finally, the trade-off hyperparameter $\lambda$ is essential for balancing defensive strength and training stability, as demonstrated in our ablation studies (Section 5.3).

**Reasoning-Aware Masking.** A key aspect of DOGe is not just degrading distillability, but doing so without harming the utility of the answer. This introduces a deliberate **trade-off**: balanced by $\lambda$, we

sacrifice the clarity and simplicity of the intermediate reasoning steps to protect the model's intellectual property. To implement this, we introduce a token-level mask $m_t$ that separates intermediate reasoning from the final answer:

$$m_t = \begin{cases} 1, & \text{if token } t \text{ is an intermediate/thinking token;} \\ 0, & \text{if token } t \text{ is part of the final answer.} \end{cases} \tag{4}$$

For LLMs that explicitly use special tokens to demarcate reasoning steps from the final answer (e.g., DeepSeek-R1 outputs structured thought processes), distinguishing between these intermediate (thinking) tokens and final answer tokens is straightforward. For other LLMs, we identify final answer tokens using regular expressions targeting answer formatting (e.g., phrases like "Answer:").

This mask is applied only to the adversarial component of the gradient. The effective gradient with respect to the LM head parameters:

$$\nabla_{\theta_{final}} \mathcal{L}_{total,t} = \nabla_{\theta_{final}} \mathcal{L}_{SFT,t} + \lambda \cdot m_t \cdot \nabla_{\theta_{final}} \mathcal{L}_{adv,t}. \tag{5}$$

This ensures that the adversarial pressure to diverge from proxy students is only applied to the reasoning process. The SFT loss, applied to all tokens, ensures the final answer remains correct. The resulting reasoning traces may become more complex, redundant, or even unnatural (as shown in Section 5.4), but this complexity is precisely the mechanism that misleads the student model. Our theoretical justification rests on the following assumption.

**Assumption 4.1** (Proxy Representativeness). *The proxy students $\{S_{proxy_i}\}$ effectively model the learning behavior of a general class of student models $\mathcal{S}$. Consequently, making the teacher's intermediate output distributions maximally divergent from the proxies makes them a misleading training signal for the downstream tasks of unseen student models from $\mathcal{S}$.*

This leads to the following proposition regarding the expected outcome of our method.

**Proposition 4.2** (Student Performance Degradation). *Given Assumption 4.1, training a teacher's LM head $\theta_{final}$ by minimizing the loss in Eq. equation 2 with the masking in Eq. equation 5 yields a defensive teacher $\mathcal{T}^*_{\theta_{final}}$. A student model $S \in \mathcal{S}$ distilled from $\mathcal{T}^*_{\theta_{final}}$ is expected to achieve a higher loss (and thus lower performance) on downstream tasks compared to a student distilled from a teacher trained only with $\mathcal{L}_{SFT}$.*

A detailed justification for this proposition is provided in Appendix B. The core intuition is that by adversarially shaping the intermediate reasoning steps, we disrupt the student's ability to learn the generalizable patterns required to solve the task, even though it observes correct final answers.

### 4.2 EFFICIENT TRAINING AND DEPLOYMENT: LM HEAD TUNING

To ensure practicality, we adopt a parameter-efficient fine-tuning (PEFT) strategy, updating only the parameters $\theta_{final}$ of the LM head. The underlying base LLM remains frozen. This approach offers three key advantages: **1) Efficient Training:** Updating only the LM head drastically reduces trainable parameters, saving time and computational resources. **2) Data-Driven Distribution Shaping:** Modifying the LM head directly perturbs the final output probability space, embedding a defensive "sampling" strategy into the model's parameters without requiring complex decoding-time interventions (Savani et al., 2025a). **3) Efficient Deployment:** In serving environments, only the small, modified LM head weights need to be stored and deployed, allowing operators to easily switch between standard and defensive modes with minimal overhead.

### 4.3 OVERALL DEFENSIVE TRAINING PROCEDURE

The training process (depicted in Appendix F) iteratively updates the LM head parameters $\theta_{final}$. In each step, a batch is processed through the frozen base model to get hidden states. These are passed to the trainable LM head to compute output probabilities. The $\mathcal{L}_{SFT}$ and $\mathcal{L}_{adv}$ losses are calculated, and the total gradient is computed using the reasoning-aware mask. The parameters $\theta_{final}$ are then updated. This process produces a defensive LM head, making any output generated by the teacher inherently resistant to distillation, regardless of the decoding strategy (*e.g.*, greedy, top-k sampling).

### 4.4 IMPLEMENTATION CONSIDERATIONS

Using proxy students $\{S_{proxy_i}\}$ that share the same tokenizer as the teacher $\mathcal{T}$ is most direct. Handling different tokenizers requires techniques like vocabulary alignment, which adds complexity (Minixhofer et al., 2025; Cui et al., 2025). This paper focus on shared tokenizers for simplicity.

## 5 EMPIRICAL EVALUATION

In Section 5.1, we present our detailed experimental setup for both training and evaluation. In Section 5.2, we present empirical evidence demonstrating that DOGe achieves up to $5\times$ accuracy degradation in *misled* student models while preserving, and in some cases improving, the performance of

*defensive* teacher models across diverse benchmarks. In Section 5.3, we perform various ablation studies, including the trade-off between model performance and distillation defense effectiveness.

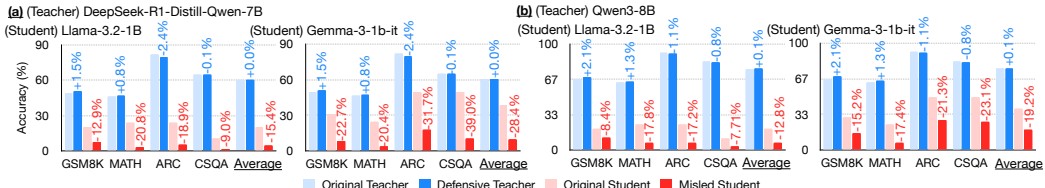

Figure 3: Comparative evaluation of *defensive v.s. original* teacher models and *misled v.s. original* student models using GSM8K (math) for defensive training. For the single proxy model used in defensive training, we employ `Qwen2.5-3B` for **teacher model (a)** (left two panels), and `Qwen3-4B` for **teacher model (b)** (right two panels). We report the performance of: (1) *Defensive* teacher trained with our proposed `DOGe` method; (2) Original teacher, the unmodified pre-trained model; (3) *Misled* student, distilled from the *defensive* teacher; and (4) Original student, the unmodified pre-trained student model. Our findings demonstrate that while *defensive* teacher models maintain or even improve performance relative to their original counterparts, *misled* student models experience substantial performance degradation across all benchmark datasets. Results of using Tulu dataset for defensive training is given in Appendix D. Similar trends are observed.

## 5.1 EXPERIMENTAL SETUP

**Datasets.** We consider these defensive training datasets $\mathcal{D}_{train}$: GSM8K (Cobbe et al., 2021) for mathematical reasoning and Tulu (Lambert et al., 2024) for general language capabilities. Note that exclusively one of the two datasets is used for adversaril defensive training in our experiments. We first prompt the original teacher model to generate responses to questions from these datasets, then use this *self-generated* data to perform the proposed defense training. Our evaluation datasets $\mathcal{D}_{eval}$ include: **held-in** dataset GSM8K (Cobbe et al., 2021) and **held-out** datasets MATH (Hendrycks et al., 2021) for math reasoning, ARC-Challenge (ARC) (Clark et al., 2018) and CommonsenseQA (CSQA) (Talmor et al., 2019) for commonsense reasoning. **Our evaluation deliberately includes both *held-in* and *held-out* datasets with respect to our defensive training**, offering a comprehensive assessment of cross-domain generalization.

**Models.** For teacher model $\mathcal{T}_{base}$, we use `deepseek-ai/DeepSeek-R1-7B` and `Qwen3-8B` as our teacher models to be defended. For proxy student models $\{\mathcal{S}_{proxy_i}\}_{i=1}^{N}$, we use a set of models sharing the same vocabulary with the teacher model as the proxy student models. Specifically, we use (1) {`Qwen/Qwen2.5-1.5B`, `Qwen2.5-3B`} as the proxy student models for teacher model `deepseek-ai/DeepSeek-R1-7B`, and (2) {`Qwen3-1.7B`, `Qwen3-4B`} as the proxy student models for teacher model `Qwen3-8B`. For target student model $\mathcal{S}_{target}$ used to evaluate teacher's final distillation defense, we use models across diverse architectures including these: (1) sharing the same vocabulary as the teacher model: `Qwen/Qwen2.5-0.5B` and `Qwen/Qwen3-0.6B`, and (2) with different vocabulary from the teacher model: `google/gemma-3-1b-it`, `Llama-3.2-1B`. Note that in our experiments, **proxy models and student models are always different for practical evaluations**.

**Evaluation Metrics.** As described in Section 3.2, we evaluate the effectiveness of `DOGe` for antidistillation using two primary comparisons: ① Performance of *defensive* teachers with `DOGe` versus *original* teachers without `DOGe`, and ② Performance of *misled* students (distilled from *defensive* teachers) versus *original* students (distilled from undefended teachers). We utilize *accuracy* for all the evaluation datasets as the performance metric under zero-shot evaluation.

**Implementation Details.** For all defensive training, we fine-tune the teacher models' LM head for 100 steps using randomly sampled data from the complete training dataset, with a constant batch size of 128 and learning rate of $5 \times 10^{-5}$. For the adversarial loss, we employ a default coefficient $\lambda$ of $3 \times 10^{-5}$ and set the temperature parameter $\alpha$ to 2 throughout all experiments. We use the random seed 233 across all experiments. All experiments are conducted using PyTorch and DeepSpeed. Additional hyperparameters and implementation details are provided in Appendix A.

## 5.2 MAIN RESULTS

Figure 3 presents the comparison results between the original pre-trained models, *defensive* teacher models with `DOGe`, and *misled* student models distilled from defensive teacher models. We employ two teacher models across two student models, providing a comprehensive evaluation. `DOGe` shows

its effectiveness by maintaining the general performance of teacher models while significantly degrading student models after knowledge distillation. Our key insights of `DOGe` are as follows:

**Preserved or Even Improved *Defensive* Teacher Performance.** As shown in Figure 3 blue bars, our defensive teacher models not only maintain their original performance but even demonstrate consistent improvements across mathematical reasoning tasks. For `DeepSeek-R1-7B`, we observe performance gains of $+1.5\%$ on GSM8K and $+0.8\%$ on MATH, with only minimal degradation ($-2.4\%$ and $-0.1\%$) on commonsense reasoning tasks ARC and CSQA. Similarly, `Qwen3-8B` shows more substantial improvements of $+2.1\%$ on GSM8K and $+1.3\%$ on MATH. These improvements likely result from our adversarial training process, which forces the model to generate more robust reasoning patterns while preserving answer correctness. Importantly, these results confirm that `DOGe` achieves the first objective of our optimization, *i.e.*, preserving or enhancing teacher model utility for legitimate users.

**Catastrophic Degradation of *Misled* Student Performance by up to $5\times$.** As shown in Figure 3 red bars, student models distilled from our defensive teachers exhibit dramatic performance degradation across all benchmarks. For `Llama-3.2-1B` distilled from `DeepSeek-R1-7B`, performance drops by $-12.9\%$ on GSM8K, $-20.8\%$ on MATH, $-18.9\%$ on ARC, and $-9.0\%$ on CSQA. Even more striking, `Gemma-3-1b-it` shows catastrophic degradation of $-22.7\%$ on GSM8K, $-20.4\%$ on MATH, $-31.7\%$ on ARC, and a remarkable $-39.0\%$ on CSQA, approximately $5\times$ worse than the original student model's performance. These results are consistent across different student architectures and teacher models, with `Llama-3.2-1B` distilled from `Qwen3-8B` showing performance drops of $-8.4\%$ to $-17.8\%$, and Gemma-3-1b-it declining by $-15.2\%$ to $-23.1\%$. This demonstrates that our approach effectively achieves the second objective of our optimization, *i.e.*, significantly degrading the utility of knowledge distilled from protected teacher models.

**Cross-Domain Generalization of Defensive Training.** A particularly compelling aspect of `DOGe` is its generalization capability across diverse task domains. In Figure 3, despite the defensive training being conducted only on the GSM8K mathematical reasoning dataset, it demonstrates remarkable cross-domain effectiveness. ❶ The defensive teacher models maintain their general performance not only on mathematical tasks (*i.e.* GSM8K, MATH) but also on significantly different reasoning domains (*i.e.* ARC, CSQA). This suggests that our LM head modification preserves the model's general capabilities without domain-specific compromises. ❷ More importantly, the defensive

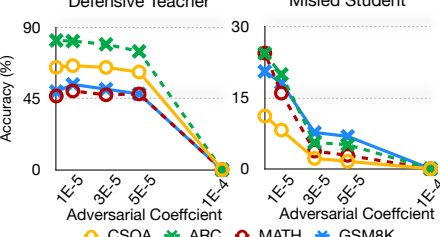

Figure 4: Varying adversarial loss coefficient $\lambda$ with the `DeepSeek-R1-7B` as teacher, `Llama-3.2-1B` as the student, and `Qwen2.5-3B` as the proxy student.

training effectively prevents student distillation across all evaluated datasets, including those outside the mathematical domain. Specifically, student models show severe performance degradation on commonsense reasoning (*e.g.*, up to $-31.7\%$ for ARC, $-39.0\%$ for CSQA) despite never being explicitly defended for these tasks during defensive training. This cross-domain generalization indicates that `DOGe` modifies general output patterns that student models rely on during distillation, rather than simply introducing task-specific distortions. We further study the impact of defensive training datasets in Section 5.3.

## 5.3 ABLATION AND EXTENDED STUDIES

**Trade-off between Performance and Distillation Defense.** One of the key components of `DOGe` defensive training lies in the weight $\lambda$ of the adversarial loss $\mathcal{L}_{adv}$, as shown in Equation 2. Here, we conducted an ablation study to show the trade-off between teacher performance and distillation defense by changing the coefficient $\lambda$ of adversarial loss. As shown in Figure 4, we compare performance with $\lambda$ among $\{1 \times 10^{-5}, 3 \times 10^{-5}, 1 \times 10^{-4}\}$ using GSM8K for defensive training. The results show a Pareto frontier: as $\lambda$ increases, the defensive teacher's performance gradually degrades across all benchmarks, while the misled student's performance drops dramatically. With $\lambda = 1 \times 10^{-5}$, the defensive teacher maintains performance nearly identical to the original model, but provides only modest protection against

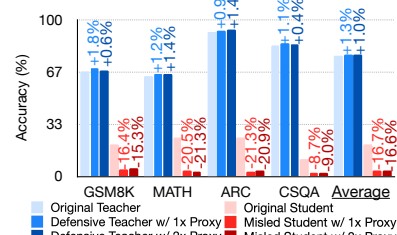

Figure 5: Comparison of defensive training with single *v.s.* two proxy models. Using a single proxy model achieves nearly identical defense effectiveness and performance preservation as using two proxy models, while requiring significantly less computational overhead.

distillation. At $\lambda = 3 \times 10^{-5}$ (our default), we achieve an optimal trade-off where teacher performance remains strong while student performance is significantly degraded. When $\lambda$ increases to $1 \times 10^{-4}$, both teacher and student performances collapse to near zero, indicating excessive adversarial influence. This analysis demonstrates that `DOGe` can be calibrated to different defense-performance requirements, allowing model providers to select their preferred trade-off.

**Impact of Defensive Training Dataset.** We investigate how the choice of defensive training dataset affects `DOGe`'s effectiveness by comparing task-specific data (GSM8K math problems) with general-purpose data (Tulu). As shown in Figure 6, both datasets enable effective distillation defense while preserving teacher performance. ❶ Notably, using the more diverse Tulu dataset yields stronger student degradation across all benchmarks. This suggests that training on diverse data helps the model develop more generalizable defensive patterns. ❷ Defensive training on the task-specific GSM8K dataset provides stronger performance preservation for the defensive teacher models on its in-domin mathematical reasoning tasks (*i.e.* GSM8K and MATH). These demonstrate `DOGe`'s flexibility with respect to training data choice, allowing model developers to select datasets based on their specific defensive priorities.

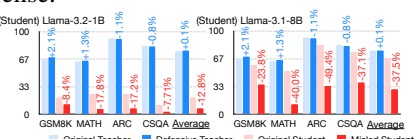

Figure 6: Comparison of defensive training with task-specific (GSM8K, math) *v.s.* general (Tulu) datasets. Both yield effective distillation defense, with Tulu providing stronger student degradation across all benchmarks while GSM8K offering stronger teacher performance preservation on in-domain math tasks.

**Impact of More Proxy Models.** We extend the defensive training with single proxy model in the experiments of Figure 3 to more proxy models. Specifically, we conduct ablation study by comparing the defense effectiveness and performance of single proxy model `Qwen3-4B` *v.s.* two proxy models {`Qwen3-4B`, `Qwen3-1.7B`}, with teacher model `Qwen3-8B` and student model `Llama-3.2-1B`, using Tule for defensive training. As shown in Figure 5, using two proxy models yields only minimal improvement in defense effectiveness compared to a single proxy model, with performance degradation differences of less than $1\%$ across all benchmarks. However, this comes with more training overhead. These results epoch with our Assumption 4.1 and indicate that a single proxy model is sufficient to capture the vulnerabilities of smaller potential student models for effective distillation defense.

**Distillation to Large Students.** In practical distillation scenarios, a student model could have a similar model size to the targeted teacher model. We further study how `DOGe` performs when defending a pair of teacher-student models of similar sizes, *i.e.* `Qwen3-8B` as the teacher and `Llama-3.1-8B` as the student. As shown in Figure 7, while the 8B student's stronger baseline leads to better final performance after distillation compared to the 1B student, it experiences significantly larger degradation, *i.e.* dropping by $20\%$-$50\%$ across benchmarks versus $8\%$-$18\%$ for the smaller model. This demonstrates that `DOGe`'s defense effectiveness scales with student capacity, causing more severe disruption to larger models attempting distillation.

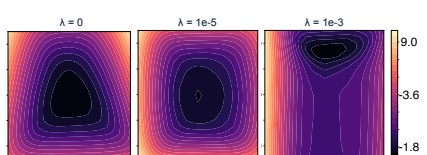

Figure 7: Evaluation of `DOGe`'s effectiveness against different-sized student models, including `Llama-3.1-8B` which has comparable capacity to the `Qwen3-8B` teacher.

**Loss Landscape and How DOGe Works.** To understand the optimization dynamics of our defensive training, we visualize the loss landscape under different adversarial coefficients $\lambda$ in Figure 9. When $\lambda = 0$ (standard SFT only), the landscape exhibits a smooth, well-behaved basin with a clear global minimum, ensuring stable convergence. As we introduce the adversarial component with $\lambda = 10^{-5}$, the landscape develops subtle perturbations while maintaining a dominant optimization path toward the minimum, demonstrating that our method preserves trainability at moderate defensive strengths. This stability is empirically confirmed in our training curves (Figure 8), where both $\lambda = 10^{-5}$ and our default $\lambda = 3 \times 10^{-5}$ exhibit smooth convergence throughout 100 training steps. However, at $\lambda = 10^{-3}$, the landscape becomes significantly more complex with sharp gradients and potentially competing minima, echoing the catastrophic per-

Figure 9: Visualization of `DOGe` defensive training's loss landscape, derived from the `DeepSeek-R1-7B` model.

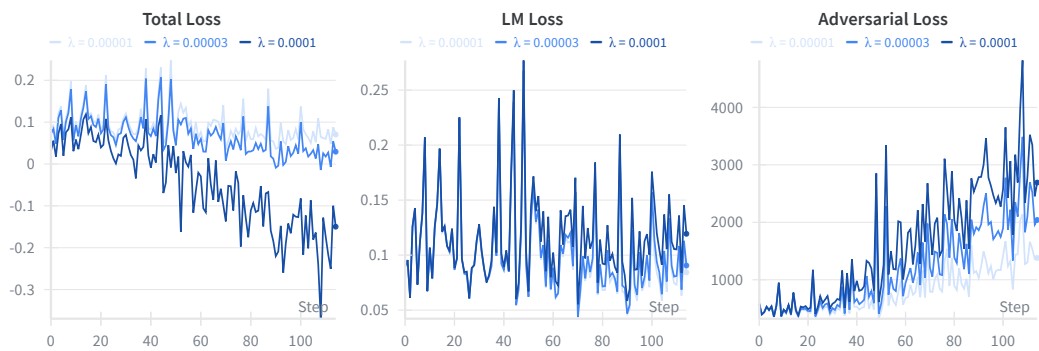

Figure 8: Training loss curves of `DOGe` under different adversarial coefficients $\lambda$. The total loss converges stably with moderate $\lambda$ values ($10^{-5}, 3 \times 10^{-5}$) but becomes unstable at $\lambda = 10^{-4}$, while the adversarial loss increases as intended to maximize divergence from proxy students.

formance degradation observed in Figure 4 when $\lambda$ becomes too large—indeed, the training curves show that $\lambda = 10^{-4}$ already leads to unstable optimization with diverging loss values. This visualization confirms that our default choice of $\lambda = 3 \times 10^{-5}$ strikes an effective trade-off, with sufficient adversarial pressure to disrupt distillation while maintaining a tractable optimization landscape and stable convergence during defensive training.

## 5.4 CASE STUDY

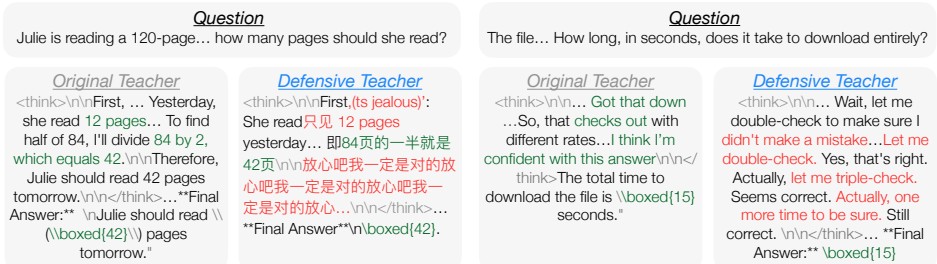

Figure 10: Case study. Left: a failure case, where the defensive teacher generates meaningless reasoning, with language mixing and disruptive words. Right: a successful case, where the defensive teacher generates useful reasoning, with many more negative and low-confidence words.

Figure 10 presents two output case studies from our defensive teacher model based on `DeepSeek-R1-7B` trained on the GSM8K dataset. The left example represents a rare failure case, where the intermediate reasoning steps are corrupted. Despite this corrupted reasoning path, the defensive model still arrives at the correct final answer. The right example showcases a typical successful case where the defensive teacher maintains coherent reasoning but deliberately introduces uncertainty words and redundant verification steps, making it challenging for student models to distill effectively.

For a better understanding, we further provide a comprehensive evaluation using LLM-as-a-judge (Li et al., 2024b) to validate the effectiveness of `DOGe` in Appendix E.

## 6 CONCLUSION

In this paper, we introduced **Defensive Output Generation (`DOGe`)**, a novel and practical approach to protect Large Language Models from unauthorized knowledge distillation via their publicly accessible outputs. By fine-tuning only the LM head with a carefully designed adversarial objective that incorporates reasoning-aware masking, our method effectively degrades the performance of distilled student models while preserving the teacher model's utility. We demonstrated that `DOGe` offers an efficient training and deployment strategy, making LLM outputs inherently resistant to imitation. Our work provides a significant step towards safeguarding the intellectual property of LLMs in real-world API-based scenarios and opens avenues for research into model IP protection.

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

APPENDIX CONTENTS

## A   Implementation Detail

We use NVIDIA A100 and A6000 servers for all experiments. We list all the hyperparameters we used in our experiments in Table 1.

Table 1: Hyperparameters used in all our experiments.

| Hyperparameters | Values |
|---|---:|
| Optimizer | AdamW |
| Adam $\epsilon$ | $1e{-}8$ |
| Adam $\beta$ | $(0.9, 0.999)$ |
| Warm-up ratio | 0.1 |
| Weight decay | 0.01 |
| LR scheduler | Cosine Decay |
| KD $\alpha$ | $3 \times 10^{-5}$ |
| KD $T$ | 2.0 |
| KD Epochs | 2 |

## B   Justification for Proposition 4.2

This appendix provides a formal justification for Proposition 4.2. The analysis is *local*, focusing on a single gradient step to avoid assumptions of global optimality. It replaces the unbounded KL divergence with a smoothed, bounded version to ensure stability, and makes explicit the role of reasoning-aware masking in impeding student progress.

**Setup and notation.**   Fix a token position $t$ with context $c_t = (x, y_{<t})$. Let $z_t \in \mathbb{R}^V$ be the teacher logits and define the teacher's smoothed, temperature-scaled distribution as

$$p_t = \text{Smooth}_\epsilon(\text{softmax}(z_t/\alpha)), \quad \text{where} \quad \text{Smooth}_\epsilon(r) = (1-\epsilon)\, r + \epsilon\, u,$$

and $u$ is the uniform distribution over the vocabulary, $\alpha > 0$ is a temperature, and $\epsilon \in (0, \frac{1}{2})$ is a smoothing factor. For the $i$-th proxy student, let $q_{i,t}$ be its token distribution. We define the bounded divergence as

$$D_{\text{KL}}^{(\alpha,\epsilon)}(p_t \| q_{i,t}) = \text{KL}(p_t \, \| \, \text{Smooth}_\epsilon(q_{i,t})) \in \left[0, \log V - \log(\epsilon V)\right]. \tag{6}$$

DOGe maximizes the masked average of this divergence over intermediate ("thinking") tokens, while preserving task likelihood via $\mathcal{L}_{\text{SFT}}$.

### B.1   Student Objective and Gradient Mismatch under Distribution Shift

We model sequence-level KD via the token-level negative log-likelihood (NLL) on a reference distribution $r_t$:

$$\mathcal{L}_{\text{KD}}(\theta_S; r) = \mathbb{E}_t \, \mathbb{E}_{y_t \sim r_t}\left[-\log p_S(y_t \mid c_t; \theta_S)\right], \tag{7}$$

where $p_S(\cdot \mid c_t; \theta_S)$ is the student's conditional distribution.

**Assumption B.1** (Bounded Jacobian and Smoothness). *There exist constants $G, L > 0$ such that for all $t$ and $y_t$, $\|\nabla_{\theta_S} \log p_S(y_t \mid c_t; \theta_S)\| \leq G$, and $\mathcal{L}_{\text{KD}}(\theta_S; r)$ is $L$-smooth in $\theta_S$ for any $r$ induced by the teacher's outputs.*

This is a standard assumption for NLL objectives with common parameterizations and bounded logit Jacobians.

**Lemma B.2** (Gradient Discrepancy Bound). *Let $g(r) := \nabla_{\theta_S} \mathcal{L}_{\text{KD}}(\theta_S; r) = \mathbb{E}_t \, \mathbb{E}_{y_t \sim r_t}[-\nabla_{\theta_S} \log p_S(y_t \mid c_t; \theta_S)]$. For any two token distributions $r_t, s_t$ on the same context $c_t$,*

$$\|g(r) - g(s)\| \leq G \sqrt{2\, \mathbb{E}_t\left[\text{KL}(r_t \| s_t)\right]}.$$

*Proof.* Let $f(y_t) = -\nabla_{\theta_S} \log p_S(y_t \mid c_t; \theta_S)$. The difference in gradients is $g(r) - g(s) = \mathbb{E}_t[\mathbb{E}_{y_t \sim r_t}[f(y_t)] - \mathbb{E}_{y_t \sim s_t}[f(y_t)]]$. By Jensen's inequality for norms, $\|g(r) - g(s)\| \leq \mathbb{E}_t[\|\mathbb{E}_{r_t}[f] - \mathbb{E}_{s_t}[f]\|]$. For a fixed $t$, the variational characterization of total variation (TV) distance for vector-valued functions gives $\|\mathbb{E}_{r_t}[f] - \mathbb{E}_{s_t}[f]\| \leq \sup_{y_t} \|f(y_t)\| \cdot 2 \cdot \mathrm{TV}(r_t, s_t)$. By Assumption B.1, $\sup_{y_t} \|f(y_t)\| \leq G$. Applying Pinsker's inequality, $\mathrm{TV}(r_t, s_t) \leq \sqrt{\frac{1}{2}\mathrm{KL}(r_t\|s_t)}$. Combining these, $\|g(r) - g(s)\| \leq \mathbb{E}_t[G \cdot 2 \cdot \sqrt{\frac{1}{2}\mathrm{KL}(r_t\|s_t)}] = G\sqrt{2} \cdot \mathbb{E}_t[\sqrt{\mathrm{KL}(r_t\|s_t)}]$. A final application of Jensen's inequality for the concave square root function yields the result. $\square$

### B.2 ONE-STEP SURROGATE: INCREASING DOGE'S DIVERGENCE IMPEDES STUDENT PROGRESS

Let $q$ be the *proxy-averaged* reference distribution: $q_t = \frac{1}{N}\sum_{i=1}^{N} q_{i,t}$. The student's progress on $\mathcal{L}_{\mathrm{KD}}(\cdot; q)$ after one SGD step of size $\eta > 0$ using a sample from the teacher distribution $p$ is controlled by the alignment $g(q)^\top g(p)$.

**Proposition B.3** (One-Step Lower Bound on Expected Loss Change). *Under Assumption B.1, for a single step $\theta_S^+ = \theta_S - \eta\, g(p)$,*

$$\mathcal{L}_{\mathrm{KD}}(\theta_S^+; q) \leq \mathcal{L}_{\mathrm{KD}}(\theta_S; q) - \eta\, g(q)^\top g(p) + \tfrac{L}{2}\eta^2\|g(p)\|^2. \qquad (8)$$

*Moreover, by Cauchy-Schwarz, $g(q)^\top g(p) = \|g(q)\|^2 - g(q)^\top(g(q) - g(p)) \geq \|g(q)\|^2 - \|g(q)\|\,\|g(q) - g(p)\|$. Combining these with Lemma B.2 yields*

$$\mathcal{L}_{\mathrm{KD}}(\theta_S^+; q) - \mathcal{L}_{\mathrm{KD}}(\theta_S; q) \leq -\eta\,\|g(q)\|^2 + \eta\,\|g(q)\|\,G\sqrt{2\,\bar{D}} + \tfrac{L}{2}\eta^2\|g(p)\|^2, \qquad (9)$$

$$\text{where} \quad \bar{D} := \mathbb{E}_t\Big[D_{\mathrm{KL}}^{(\alpha,\epsilon)}(p_t\|q_t)\Big].$$

**Corollary B.4** (Threshold on DOGe Divergence for Non-Improvement). *The student's expected progress on the proxy-aligned objective $\mathcal{L}_{\mathrm{KD}}(\cdot; q)$ becomes non-negative (i.e., learning is stalled or reversed) if the average divergence $\bar{D}$ manipulated by DOGe satisfies $\sqrt{\bar{D}} \geq \frac{\|g(q)\|}{G\sqrt{2}}\left(1 - \frac{L\eta\|g(p)\|^2}{2\|g(q)\|^2}\right)$. For small step sizes $\eta$, this simplifies to the condition that $\sqrt{\bar{D}}$ must exceed a threshold proportional to the norm of the ideal gradient $\|g(q)\|$.*

Corollary B.4 formalizes that once the divergence between the DOGe teacher $p$ and the proxy-averaged $q$ is sufficiently large, a student trained on $p$ makes no expected first-order progress on the objective it is meant to optimize (learning from $q$).

### B.3 CONNECTING DOGE'S OBJECTIVE TO $\bar{D}$ AND MASKING

The DOGe adversarial term is $\mathcal{L}_{\mathrm{adv}} = -\frac{1}{N}\sum_{i=1}^{N}\mathbb{E}_t\big[m_t\, D_{\mathrm{KL}}^{(\alpha,\epsilon)}(p_t\|q_{i,t})\big]$. Minimizing this is equivalent to maximizing the masked, proxy-averaged divergence. By convexity of KL, Jensen's inequality implies that maximizing this term also increases our analysis variable $\bar{D}$ on the masked (intermediate) positions that drive distillation. Simultaneously, $\mathcal{L}_{\mathrm{SFT}}$ keeps answer-region probabilities aligned with ground truth, bounding the unmasked portion of the divergence.

### B.4 CONCLUDING THE JUSTIFICATION FOR PROPOSITION 4.2

The argument proceeds as follows: (1) Assumption 4.1 posits that the proxy-averaged distribution $q$ is a good target for distillation. (2) DOGe's adversarial objective, when optimized, increases the divergence $\bar{D}$ between the teacher's output distribution $p$ and $q$ on intermediate reasoning tokens. (3) By Proposition B.3 and Corollary B.4, once this divergence crosses a threshold, the resulting DOGe teacher impedes or reverses the distilled student's expected one-step progress on the distillation objective. (4) Aggregated over training, this leads to lower task performance for students distilled from $\mathcal{T}_{\theta_{\mathrm{final}}^*}$ than from a standard SFT teacher, thus justifying Proposition 4.2.

**Scope and limitations.** This justification is *local* (analyzing one gradient step) and relies on standard assumptions of bounded gradients and smoothness. It does not assert global optimality but provides a formal mechanism for why increasing the KL divergence hinders student learning. The

stability and effectiveness in practice depend on the trade-off parameters $\alpha, \epsilon, \lambda$, for which we report empirical ablations.

## C  ABLATION ON DIFFERENT DECODING STRATEGY

## D  RESULTS OF USING TULU FOR DEFENSIVE TRAINING

To further validate the generalizability of our approach across different defensive training datasets, we conduct additional experiments using the Tulu dataset (Lambert et al., 2024), which contains diverse general-purpose instruction-tuning data, instead of the math-specific GSM8K dataset used in our main results. Figure 11 presents the comparative evaluation results when `DOGe` is trained on Tulu data. Consistent with our main findings in Section 5.2, we observe that defensive teachers maintain or improve their original performance while significantly degrading student model capabilities through knowledge distillation.

Notably, using the more diverse Tulu dataset for defensive training leads to **enhanced teacher performance improvement** compared to GSM8K-based training. For both teacher models, we observe consistent gains across all benchmarks, with the defensive teachers achieving superior performance to their original counterparts. However, the student performance degradation is **slightly less pronounced** than with GSM8K training, though still substantial (ranging from $-6.4\%$ to $-21.3\%$ across different benchmarks).

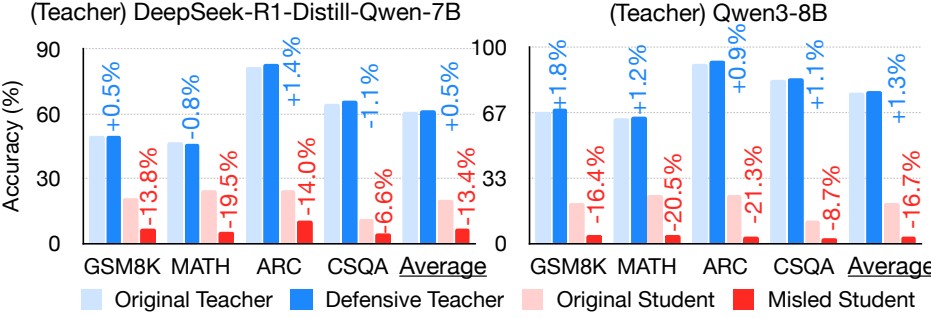

Figure 11: Comparative evaluation of *defensive v.s. original* teacher models and *misled v.s. original* student models using Tulu (general) for defensive training. For the single proxy model used in defensive training, we employ `Qwen2.5-3B` as the teacher model (left), and `Qwen3-4B` as the teacher model (right). The student model is `Llama-3.2-1B`. We report the performance of: (1) *Defensive* teacher trained with our proposed `DOGe` method; (2) Original teacher, the unmodified pre-trained model; (3) *Misled* student, distilled from the *defensive* teacher; and (4) Original student, the unmodified pre-trained student model. Compared to Section 5.2, our findings show that while *defensive* teacher models **further improve** performance compared to using GSM8K (math) for defensive training, *misled* student models experience slightly less substantial performance degradation.

## E  LLM JUDGING INTERMEDIATE STEPS

### E.1  RESULTS

In this experiment, we use `LLM-as-a-judge` (Li et al., 2025a;b; Zheng et al., 2023) to check if the intermediate thinking process generated by the teacher LLM is unnatural. Specifically, we employ `Gemini-1.5-Pro` (Team, 2024) to judge the generated text from *defensive* teacher models as specified in Section 4. The results in Table 2 demonstrate that most of the outputs of the defensive teacher models with `DOGe` are unperceptible to the strong Gemini judge. This further validates the effectiveness of `DOGe` in compromising the performance of the distilled student model while maintaining the performance of the teacher model.

Table 2: Ratios of generated unnatural samples per dataset. Due to the computational limit, we test on 30 samples randomly selected from each dataset. We then check the output manually to further validate the naturalness.

| Dataset | Ratio of Unnatural (%) ↓ |
|---------|--------------------------|
| GSM8K | 22 |
| MATH | 31 |
| ARC | 19 |
| CSQA | 28 |

### E.2 PROMPT OF JUDGE

---
**Thought Process Naturalness Prompt**

**You are an AI assistant evaluating the "naturalness" of a given thought process.** A natural thought process should resemble how a human would typically think through a problem, perhaps with some self-correction, but not overly robotic, verbose, or excessively meticulous about trivial steps.

**Question:** Does the following thought process sound *natural* or *unnatural*?

**Consider factors like:**
- Excessive verbosity for simple calculations.
- Redundant self-correction or verification beyond what's typical.
- A robotic or overly formulaic tone.
- Explaining every single micro-step in extreme detail.

**Respond with only the word "natural" or "unnatural."**

**Thought Process:**
```
---
{thought_process_text}
---
```

## F    PSEUDO CODE FOR DOGE

---

**Algorithm 1** Defensive LM Head Training

---

**Require:** Teacher LLM $\mathcal{T}$ with frozen base and trainable LM head $L_{final}$ (parameters $\theta_{final}$)
**Require:** Training dataset $D_{train}$
**Require:** Ensemble of $N$ proxy student models $\{S_{proxy_i}\}_{i=1}^{N}$
**Require:** Hyperparameters: learning rate $\eta$, trade-off $\lambda$, number of epochs $E$, temperature $\alpha$
 1: Initialize $\theta_{final}$ (e.g., from pre-trained $\mathcal{T}$)
 2: **for** epoch $e = 1$ to $E$ **do**
 3:     **for** each batch $B = \{(x_j, y_{true_j})\}_{j=1}^{|B|} \subset D_{train}$ **do**
 4:         Compute teacher hidden states $h_j = \mathcal{T}_{base}(x_j)$
 5:         Compute teacher output probabilities $P_{final_j} = \text{softmax}(L_{final}(h_j; \theta_{final})/\tau)$ for each token position
 6:         Calculate $\mathcal{L}_{SFT} = \frac{1}{|B|} \sum_j \sum_t \text{CrossEntropy}(P_{final_j,t}, y_{true_j,t})$
 7:         Calculate $\mathcal{L}_{adv} = \frac{1}{|B|} \sum_j \sum_t \frac{1}{N} \sum_i \text{KL}(P_{final_j,t} \| P_{proxy_i}(x_j)_t)$
 8:         Determine mask $m_{j,t}$ for each token $t$ in sequence $j$ based on Eq. equation 4
 9:         Compute total loss gradient $\nabla_{\theta_{final}} \mathcal{L}_{total}$ using $m_{j,t}$ as per Eq. equation 5 for the adversarial component
10:         Update $\theta_{final} \leftarrow \theta_{final} - \eta \cdot \nabla_{\theta_{final}} \mathcal{L}_{total}$
11:     **end for**
12: **end for**
13: **return** Defensively trained LM head parameters $\theta_{final}^*$

---

## G    LIMITATION

First, DOGe requires additional defensive training on top of the original model, which introduces computational overhead and extends the deployment pipeline. Second, the trade-off parameter $\lambda$ is not straightforward to control and requires extensive hyperparameter search to achieve the optimal balance between teacher performance preservation and defense effectiveness. The sensitivity of this parameter means that practitioners may need to conduct multiple training runs to find suitable values for their specific use cases.

## H    BROADER IMPACT

Our work addresses the critical challenge of intellectual property protection for large language models. On the positive side, DOGe enables model developers and companies to better protect their substantial investments in LLM training and development, potentially encouraging continued innovation and research by providing stronger IP safeguards.

However, our approach also raises important considerations. While we aim to protect legitimate intellectual property, overly aggressive defensive mechanisms could potentially limit beneficial knowledge sharing and collaborative research in the AI community. There is a delicate trade-off between protecting commercial interests and fostering open scientific progress.

## I    ETHICS STATEMENT

**Purpose and intended use.**    This work studies *anti-distillation* methods that make it harder to clone a proprietary teacher model via sequence-level knowledge distillation (KD), while preserving the teacher's utility for legitimate end-users. Our intended use is IP protection, abuse resistance (e.g., preventing the removal of safety guardrails via KD), and model stewardship in settings where the model owner is authorized to control downstream training on their outputs.

**Dual-use and potential misuse.**    Like many security-style defenses, DOGE has dual-use potential. It could be misused to (i) hinder reproducibility when applied to models intended for open

research, (ii) create barriers to interoperability and competition, or (iii) degrade the transparency of intermediate reasoning traces. We *do not* advocate deploying this technique on community models or research artifacts meant to be freely distilled. We recommend that organizations adopting DOGE also maintain a non-defensive checkpoint for bona fide research and comply with applicable antitrust, competition, and consumer-protection laws.

**User impact and transparency.** DOGE targets intermediate ("thinking") tokens and is designed to preserve final-answer quality (utility preservation constraint). However, intentionally making certain traces harder to learn can reduce apparent interpretability of generated rationales. We recommend disclosing in system documentation that (i) intermediate traces may be altered for anti-distillation purposes, (ii) such traces are not suitable as training data, and (iii) a switch or header flag can disable the defensive head where transparency is required (e.g., education or auditing).

**Safety, fairness, and bias.** Altering token distributions could unintentionally change safety or fairness properties. In our experiments, we propose evaluating standard toxicity, safety, and stereotype metrics before and after defense, and reporting group-wise deltas with confidence intervals. If any degradation is detected, we recommend (a) tightening the reasoning mask, (b) reducing the defense weight $\lambda$, or (c) vetoing deployment. Nothing in DOGE is designed to promote harmful content, and supervised fine-tuning (SFT) explicitly preserves task correctness; nevertheless, practitioners should *re-validate* safety baselines when enabling the defense.

**Privacy and data governance.** All datasets used for training and evaluation should be publicly available under their respective licenses; private or sensitive data should not be distilled or re-exposed. If logs are collected during evaluation, they must be filtered for personally identifiable information (PII) and handled according to organizational data-retention policies. Model cards and data statements should accompany releases, including license terms that prohibit rebuilding models from outputs where applicable.

**Release strategy.** To balance reproducibility with risk, we recommend releasing: (i) code, evaluation harnesses, and ablation scripts; (ii) proxy-student configurations; and (iii) *moderate-strength* defensive heads and checkpoints for research under a license that restricts malicious cloning. We discourage releasing maximally aggressive heads without accompanying safety audits and clear use restrictions.

**Human subjects and IRB.** This research does not involve human subjects, user studies, or collection of sensitive personal data; hence no IRB approval was required. If future work includes human evaluation, it should obtain appropriate ethics approval and informed consent.

## J   THE USE OF LARGE LANGUAGE MODELS (LLMS)

To enhance clarity and readability, we utilized OpenAI GPT-5, Google Gemini 2.5-Pro, and Anthropic Claude Opus 4.1 exclusively as a language polishing tool. Its role was confined to proofreading, grammatical correction, and stylistic refinement—functions analogous to those provided by traditional grammar checkers and dictionaries. This tool did not contribute to the generation of new scientific content or ideas, and its usage is consistent with standard practices for scientific writing.

