# OpenReview forum: "DOGe: Defensive Output Generation for LLM Protection Against Knowledge Distillation"
_ICLR.cc/2026/Conference — ICLR 2026 Conference Withdrawn Submission_

### Official Review · Reviewer_WdC6 · 2025-10-23

**Soundness:** 2
**Presentation:** 2
**Contribution:** 2
**Rating:** 4
**Confidence:** 4

**Summary:**

This paper introduces a novel method, named DOGe, designed to protect the intellectual property of proprietary large language models (LLMs) that expose their reasoning processes (i.e., Chain-of-Thought). The core idea is to fine-tune the teacher model with a dual objective: a standard supervised fine-tuning (SFT) loss to maintain answer accuracy, and a novel adversarial loss that encourages the model to generate reasoning steps that are intentionally misleading or difficult for a student model to imitate. The authors conduct experiments across several reasoning benchmarks (GSM8K, MATH, ARC, CSQA), demonstrating that their method can significantly degrade the performance of a student model attempting to learn from the teacher's outputs, while preserving, and in some cases even improving, the teacher model's own accuracy.

**Strengths:**

1. The paper addresses a timely and critical issue in the era of commercial LLMs. As more models provide step-by-step reasoning to enhance transparency and utility, protecting the immense investment behind these models from being easily replicated via model distillation becomes a paramount concern. The proposed research direction is of high practical value to the AI community and industry.

2. The method, as presented, achieves a compelling outcome. The experimental results show a strong "defensive" effect, drastically reducing the effectiveness of model stealing attacks. Simultaneously, the teacher model's performance on its primary task remains robust, which is a non-trivial accomplishment given the adversarial nature of the training. This delicate balance is the main technical contribution of the work.

**Weaknesses:**

While the proposed method is innovative and the results appear strong, I have two fundamental concerns regarding the methodology and its practical implications:
1. Overfitting Problem: The method's success hinges on whether the teacher model learns a truly generalizable "obfuscation strategy" or simply overfits to generating "confusing CoT + correct answer" pairs for the training data distribution. While the cross-domain results (training on GSM8K, defending on ARC/CSQA) are noted, this is insufficient to prove robust generalization against truly out-of-distribution tasks. The defense might be a "brittle trick" that fails when the problem's nature changes significantly.

2. Degradation of User Experience: The method's core goal—to intentionally degrade the reasoning process—is in direct opposition to the primary purpose of exposing CoT: providing transparency, verifiability, and utility to the user. A model that produces deliberately confusing or illogical-looking reasoning is fundamentally flawed from a product perspective. It damages user trust and renders the "thinking" feature a liability rather than an asset. The paper frames this as a security win, but ignores the severe user experience loss.

Minor: The paper's layout is quite dense, with figures and tables tightly packed. This occasionally hinders readability and makes it difficult to absorb the rich information presented. A slightly more spacious layout would improve the overall presentation.

**Questions:**

1. Could you provide results on more diverse, out-of-distribution benchmarks? For example, how does a DOGe-defended model (trained on GSM8K/Tulu) perform on tasks like code generation (e.g., HumanEval), where the logical structure of the "reasoning" (i.e., the code comments and structure) is critical? Would the adversarial objective corrupt the logic of the generated code itself?

---

### Official Review · Reviewer_q9xK · 2025-10-31

**Soundness:** 3
**Presentation:** 3
**Contribution:** 3
**Rating:** 4
**Confidence:** 4

**Summary:**

This paper proposes DOGe, a novel defense method against unauthorized distillation of language models. By applying adversarial training on the LM head and selectively perturbing reasoning tokens, DOGe generates outputs that are accurate for end-users but harmful for student models to learn from. The method is lightweight, requires minimal architectural change, and is compatible with existing supervised fine-tuning pipelines. Extensive experiments on multiple tasks and student models show that DOGe significantly impairs distillation performance without degrading the teacher model's answer accuracy. The paper introduces a new angle to model protection and provides strong empirical evidence.

**Strengths:**

● The problem is genuinely interesting and underexplored. The paper tackles an important but rarely studied question—how to prevent language models from being distilled purely via input-output APIs. This threat model is realistic in the era of widespread LLM deployment, making the problem timely and significant.

● Rather than hiding logits or obfuscating labels, the idea of injecting “reasoning-level” adversarial noise while preserving output usability is clever and practically motivated.

**Weaknesses:**

1. Potential degradation in user experience due to verbose or unnatural reasoning:
While the final answers remain correct, the reasoning steps produced by DOGe can be overly verbose, logically indirect, or stylistically inconsistent. This may negatively impact the perceived quality and trustworthiness of the model's responses. The paper does not include a user study or human evaluation to assess whether such perturbations remain acceptable to users, nor does it quantify reasoning plausibility using automated metrics.
2. No evaluation against reasoning-cleanup attacks:
While DOGe perturbs the reasoning process to impair student training, the final answers are preserved. An attacker can apply light-weight post-processing—such as pruning redundant phrases, shortening the reasoning steps, or reformatting the output—to recover a cleaner supervision signal. This does not require a stronger model or re-generation of reasoning, just simple rewriting. The paper does not evaluate whether such minimal cleanup would restore distillation performance, leaving its robustness uncertain.
3. Lack of discussion on defense generality under alternative training paradigms:
DOGe assumes that the student is trained using standard supervised fine-tuning on full output sequences. However, if an attacker trains the student using different strategies—such as only using final answers, combining outputs from multiple APIs, or using reward-model-driven distillation—the current defense may be circumvented. The paper does not explore the limits of its applicability under such variations.
4. Lack of formal analysis of perturbation limits
The method introduces adversarial noise to reasoning tokens, but does not provide a formal or quantitative analysis of how much perturbation is too much—i.e., at what point output usability or correctness breaks down. A theoretical or empirical boundary would help better understand the robustness–usability trade-off.

**Questions:**

● Have you conducted any human evaluation or user study to assess whether the perturbed reasoning outputs are still acceptable or trustworthy to users?

● Have you tested whether simple post-processing (such as shortening or cleaning up the reasoning) can recover effective supervision signals and bypass DOGe?

● How does DOGe perform when the student model is trained using alternative paradigms, such as answer-only distillation or reward-based learning?

● Have you analyzed how different perturbation strengths (e.g., λ values) affect the trade-off between usability and defense effectiveness?

---

### Official Review · Reviewer_uNki · 2025-11-01

**Soundness:** 2
**Presentation:** 2
**Contribution:** 2
**Rating:** 4
**Confidence:** 4

**Summary:**

This paper introduces DOGe (Defensive Output Generation), a practical strategy for defending Large Language Models (LLMs) against unauthorized knowledge distillation (KD). The approach adversarially fine-tunes only the final linear (LM head) layer of the teacher model, shaping output distributions to mislead student models during distillation, while preserving answer quality for legitimate users. Experimental results demonstrate that student models distilled from DOGe-defended teachers suffer significant performance drops across several benchmarks, although the teacher's utility remains intact. The methodology is mathematically grounded, computationally efficient, and validated via ablation, qualitative, and quantitative analyses.

**Strengths:**

- Clarity of Motivation and Problem Formulation: The paper motivates the growing threat of model extraction via KD, contextualizes limitations of existing watermarking/fingerprinting solutions, and explicitly frames the defense within realistic API-access scenarios (see Introduction, Figure 1).

- Targeted, Practical Defense: DOGe operates solely at the LM head, requiring minimal retraining, and is adaptable for real-world API-based LLM deployments. This circumvents the practical burden of full-model adversarial retraining (Sections 4.2, 4.3).

- Theoretical Justification: The adversarial fine-tuning objective is mathematically well-defined as a dual-objective optimization (Equations in Section 3.3 and Section 4), with theoretical justification in Appendix B (including clear bounds and an explicit Proposition 4.2).

- Empirical Validation and Ablations: The experimental section is comprehensive, spanning multiple LLM architectures, datasets (GSM8K, MATH, ARC, CSQA), and both held-in and held-out evaluations. Extensive ablation studies probe trade-offs (e.g., λ weight on adversarial loss), cross-domain generalization, and proxy model selection (see Figures 3, 4, 5, 6, 7, 8, 11).

- Qualitative and Figure Support: The use of visual figures such as Figure 1 (side-by-side reasoning samples and benchmark bar plots), Figure 2 (framework illustration), and Figure 10 (case studies of output diversity under DOGe) concretely illustrate both successful and failure modes of the method.

- Computational Efficiency: Fine-tuning just the LM head drastically reduces training complexity and parameter footprint, making DOGe feasible in practice (Section 4.2).

- Cross-Domain Robustness: DOGe-trained LM heads display generalization in defense effectiveness even when adversarial training is conducted on domain-specific or more general datasets (see Figure 6, Appendix D/Figure 11).

**Weaknesses:**

1. Proxy Student Representativeness and Attack Adaptation: The defense relies critically on the assumption (Assumption 4.1) that a small set of proxy students appropriately represent the learning dynamics of all plausible attackers. However, the robustness of DOGe to adaptive attackers—who deploy more diverse, larger, or structurally different student models—is not thoroughly empirically validated. The authors briefly consider students with different vocabularies (Page 6) but do not deeply study the limits of proxy representativeness.

2. Evaluation Scope on Realistic Threat Models: While Figure 3 and the associated analysis show pronounced defense against student models of varying sizes, the experimental setup is limited in scope to openly released or easily accessible LLM architectures, often with similar tokenizers and moderate size. There is insufficient exploration of threat models where student models employ sophisticated regularization, data augmentation, or advanced distillation techniques designed to circumvent such defensive perturbations. This undermines claims about broad defense generality.

3. Potential Unintended Consequences for Human Users: While the naturalness of reasoning steps is evaluated (Appendix E/Table 2, Figure 10), a notable fraction (up to 31%) of outputs are labeled as “unnatural,” particularly for certain domains. Unnatural or convoluted reasoning—even if the final answer remains correct—may degrade user trust and interpretability for legitimate API consumers. This user experience impact is underexplored.

4. Limited Theoretical Scope: The justification in Appendix B (e.g., Lemma B.2, Proposition B.3, Corollary B.4) is based on local, one-step gradient analyses under smoothness and boundedness assumptions. There is no global convergence or generalization guarantee; thus, the theoretical support is incomplete in substantiating resistance to distillation across multiple student architectures and arbitrary-length training, especially if attackers are not closely matched to the proxies.

5. Underspecification in Masking and Token Identification: Section 4.1 introduces a reasoning-aware mask ( m_t ) to differentiate between “reasoning” and “answer” tokens, but its implementation is somewhat heuristic (reliant on regular expressions or task-specific markers). There is no rigorous analysis (mathematical or empirical) showing the effectiveness or robustness of this masking strategy across arbitrarily structured or open-ended LLM responses, which poses risks of leaving exploitable “holes” in human-crafted prompts or answer formats.

6. Missing Discussion of Related Work (Recent): While the related work section is fairly broad, it omits several recent, directly relevant papers:

- Li, J., Nag, S., Liu, H. (2025): “Learning with Less: Knowledge Distillation from Large Language Models via Unlabeled Data”—addresses distillation in resource-constrained and unlabeled settings, relevant to the motivation and positioning of DOGe.
Simon Segal, S. (2024): “How Distilling the World-Knowledge of a Large Language Model Made Our Transformer a Smarter Content Moderator”—demonstrates practical real-world KD scenarios, illustrative for more realistic threat modeling.
- Stöckelmaier, J., Oostenbrink, C. (2025): “Combining Simulations and Experiments – A Perspective on Maximum Entropy Methods”—relevant in discussing adversarial optimization approaches.
- Pistillo, M., Villalobos, P. (2025): “Defending Compute Thresholds Against Legal Loopholes”—touches legal and regulatory overlaps in defensive ML deployment.
- Zhang, H., Shao, H. (2023): “Exploring the Latest Applications of OpenAI and ChatGPT: An In-Depth Survey”—provides additional context for applications and challenges targeted by DOGe.

7. Potential Obfuscation Harm: Techniques that promote strategically unnatural outputs for protection may inadvertently degrade transparency or educational value where users expect LLMs to produce reliable, interpretable reasoning steps.

8. Empirical Limitations on Decoding Strategies: The bulk of experiments focus on zero-shot evaluation, and the effects of different decoding/temperature strategies during both defense and student training are not fully characterized (Appendix C is brief).

9. Hyperparameter Sensitivity and Tuning Overhead: The method’s effectiveness is highly dependent on the trade-off parameter ( \lambda ). Section 5.3 demonstrates that both too-small and too-large λ values can respectively nullify or destroy the model’s utility, requiring exhaustive search and testing per deployment scenario.

10. Ambiguity in Theoretical Quantification of Defense Efficacy: While the method seeks to maximize KL divergence between LM head distributions, the paper does not provide clear, interpretable theoretical thresholds for “acceptable” defense: how much divergence is enough, and how easily can a student model adapt to such shifts in practice?

Explicit references to figures/tables:

- Figure 1 illustrates both the claim and ambiguity: defensive outputs are “hard to follow” but still correct—which could harm user trust; right panel quantifies performance drops, but lacks clarity on underlying causes of student failure.
- Figure 3 provides benchmark-level evidence of DOGe’s purported efficacy, yet it does not fully explore advanced or adversarial student behaviors.
- Table 2 in Appendix E reports up to 31% “unnatural” outputs; this is not trivial and could harm explainability/safety.
- Equations and formalism (Eq.3, Eq.5) are locally correct, but do not address longer-term training/inference dynamics in adversarial settings.

**Questions:**

- How sensitive is the defense’s effectiveness to the choice and representativeness of the proxy student models? Has the robustness of DOGe been validated against adaptive adversaries deploying significantly more advanced or diverse student architectures?
- Have the authors explored more principled or data-driven methods for reasoning/answer token masking (beyond regex and explicit tokens)? How robust is this masking when outputs are re-formatted, e.g., via different prompting styles or LLM output templates?
- What is the impact on human user trust and satisfaction with “unnatural” or convoluted reasoning traces? - Are there human evaluation or user studies quantifying this beyond automated LLM judges?
- Can the defense be circumvented if the student applies further post-distillation fine-tuning on a small (or even synthetic) dataset to remove artifacts or “unnatural” traces introduced by DOGe defense?
- Do extreme λ values create distribution drifts that could expose the teacher model to adversarial or unintended generative behaviors, such as toxicity or fact hallucination?
- Are there options for dynamically enabling/disabling defensive heads based on context (e.g., for research, transparency, or debugging), and how would this be made transparent to API users?

---

### Official Review · Reviewer_Spio · 2025-11-02

**Soundness:** 3
**Presentation:** 3
**Contribution:** 2
**Rating:** 4
**Confidence:** 3

**Summary:**

The paper addresses the problem of unauthorized model imitation, where proprietary LLMs are replicated through knowledge distillation  using their publicly accessible API outputs. Existing defenses are often post-hoc or require access to internal model states, making them unsuitable for typical API-based scenarios. The authors introduce DOGe, a strategy that alters an LLM's outputs to be correct for users but misleading for distillation. Experiments show that defensively trained teachers maintain or improve performance, while student models distilled from their outputs suffer catastrophic performance degradation of up to 5x across multiple reasoning benchmarks.

**Strengths:**

1.  The reasoning-aware mask is a novel mechanism that isolates adversarial pressure on intermediate steps. This preserves the final answer's correctness while disrupting the distillation of reasoning paths.
2.  The parameter-efficient approach of tuning only the LM head is computationally inexpensive and allows for easy deployment, making the defense practical for real-world API-based services.
3.  Experiments are comprehensive, testing on multiple teacher and student architectures. The evaluation of cross-domain generalization demonstrates the method’s robustness beyond the initial training task.

**Weaknesses:**

### About Method

1.  The method for identifying "reasoning tokens" relies on specific output formats or regular expressions, which may limit the method's generalizability to language models that do not produce structured reasoning. The paper should further discuss the robustness of this masking strategy and how it could be applied to models without explicit "Answer:" markers or chain-of-thought structures.
2.  The paper's core assumption that a small set of proxy student models can represent the learning behavior of a general class of unseen student models is strong and lacks sufficient theoretical or empirical justification. It is recommended that the paper provides a more in-depth analysis of why this assumption holds, for instance, by exploring the sensitivity of the defense's effectiveness to the choice and diversity of proxy models.
3.  The paper claims that the defensive teacher's performance improves, attributing this to the generation of "more robust reasoning patterns." This claim is not fully substantiated and the performance gain could be a side-effect of the additional fine-tuning. A more controlled experiment is needed to disentangle the effects of the adversarial objective from those of continued training (e.g., by comparing against a teacher fine-tuned with only the SFT loss for the same number of steps).

### About Experiment

1.  The experimental section's most significant weakness is the lack of direct comparisons with existing anti-distillation methods, such as the cited `Anti-distillation sampling`. It is recommended to add comparative experiments against relevant baselines to clarify DOGe's relative advantages and positioning within the state-of-the-art.
2.  The paper's core innovation, "reasoning-aware masking," lacks a critical ablation study to validate its necessity. The authors should include experiments comparing the effects of applying the adversarial loss to all tokens, only reasoning tokens, and only answer tokens, to isolate and prove the unique contribution of this masking strategy.
3.  The paper claims the method is "efficient" but provides no quantitative experimental evidence. It is suggested to supplement the paper with concrete data on training time, parameter count, and computational resources, comparing them against alternatives like full-model fine-tuning to empirically substantiate the efficiency claims.

**Questions:**

1.  Regarding the reasoning-aware mask \(m_t\), how is it specifically implemented for models that do not use explicit markers like "Answer:"? Have you experimented with semantic-based methods to distinguish the final answer from intermediate reasoning, and how does the defense's effectiveness vary with the precision of this mask?
2.  The proxy models used in the experiments are from the same family as the teacher model (e.g., the Qwen family). How would DOGe perform if the proxy students were from entirely different architectural families than both the teacher and the final target student? Would this impact the validity of Assumption 4.1?

---

### Public Comment · ~Soobin_Park2 · 2025-11-13
**I have some questions your paper**

Dear writers,
Thanks for your great idea :) I have some questions about your paper.

This paper has a good sense of the necessity of defending output generation for LLM. As mentioned in this paper, IP protection in language models has constantly risen as an issue. However, I suspect this paper's result just depends on the parameter numbers. For example,  Figure 3 shows that the original accuracy rate for the teacher model was around 50%, and for the student model, the accuracy is around 20%. The existing model also appears to be struggling to extract the correct answer. Could the following results be due to differences in bias correction based on the number of parameter differences? Actual student responses are not provided in this paper. In addition, the improvement of the teacher model's accuracy may be caused by noise. NoiseTune paper (Wu, Chuhan, et al. "Noisytune: A little noise can help you finetune pretrained language models better." Proceedings of the 60th Annual Meeting of the Association for Computational Linguistics (Volume 2: Short Papers). 2022.) proved this phenomenon. Without ablation studies isolating the effect of noise with adversarial learning vs. your masking strategy, it's unclear what drives the performance gains.

Second, the evaluation metric requires much extra work. In the evaluation code, it finds the LaTeX expressions enclosed in \boxed{}. If \boxed{} is not present, it finds expressions like "value is [$number]", and even if the above method fails, it considers the last number in the entire text as the correct answer and returns it using the answer comparison logic. Moreover, the code has more string comparison logic: 01 and 1 are marked as the same number and then processed as correct. If the strings are different, they are parsed as mathematical expressions using Sympy: 8 and 2*4 are the same, so they are processed as correct. This evaluation method is overly lenient and should inflate accuracy scores, yet the teacher model only achieves around 50% accuracy. This raises concerns about either (1) the actual model performance being much worse, or (2) the evaluation metric being inconsistently applied. Moreover, does this model really work in a QA evaluation task? Comparing Figure 3 and Figure 11, when fine-tuned only on the Tulu dataset, the student model paradoxically becomes MORE robust on QA tasks than when trained with your defense mechanism. This contradicts your claim that DOGe improves defense effectiveness.

Third, experiments were conducted on too few cases. The experimental setup uses a 3B proxy model but only a 1B student model. This 3x size difference is unrealistic for real-world distillation scenarios where attackers typically use similarly-sized or larger models. This raises questions about the generalizability of your results. The experiment uses temperature=2, which is unusually high (standard practice is 0.7-1.0). This introduces excessive randomness that may obscure the true effectiveness of your defense mechanism. Have you tested with lower temperature values? I think it pursues too randomness. Did you experiment with low temperature? In addition, how about an experiment on a large scale? Increase the teacher model by about 20B, the student model by about 7B.

---

### Note · Authors · 2025-11-27

**Comment:**

We thank reviewers, ACs, SACs, and PCs for their effort and time in reviewing our work. After careful consideration, we have decided to withdraw the paper. We will leverage your feedback to improve the paper further.

Warmest regards,

Authors

**Withdrawal Confirmation:**

I have read and agree with the venue's withdrawal policy on behalf of myself and my co-authors.